# An Investigation into Compound Likelihood Ratios for Forensic DNA Mixtures

**DOI:** 10.3390/genes14030714

**Published:** 2023-03-14

**Authors:** Richard Wivell, Hannah Kelly, Jason Kokoszka, Jace Daniels, Laura Dickson, John Buckleton, Jo-Anne Bright

**Affiliations:** 1Institute of Environmental Science and Research Limited, Private Bag 92012, Auckland, New Zealand; 2Alabama Department of Forensic Sciences, 1 Forensic Drive, Mobile, AL 36617, USA; 3Washoe County Sherriff’s Office, Forensic Science Division, 911 Parr Blvd, Reno, NV 89512, USA; 4Department of Statistics, University of Auckland, Private Bag 92019, Auckland, New Zealand

**Keywords:** forensic DNA analysis, mixtures, propositions, likelihood ratios

## Abstract

Simple propositions are defined as those with one POI and the remaining contributors unknown under *H_p_* and all unknown contributors under *H_a_*. Conditional propositions are defined as those with one POI, one or more assumed contributors, and the remaining contributors (if any) unknown under *H_p_*, and the assumed contributor(s) and N unknown contributors under *H_a_*. In this study, compound propositions are those with multiple POI and the remaining contributors unknown under *H_p_* and all unknown contributors under Ha. We study the performance of these three proposition sets on thirty-two samples (two laboratories × four NOCs × four mixtures) consisting of four mixtures, each with N = 2, N = 3, N = 4, and N = 5 contributors using the probabilistic genotyping software, STRmix™. In this study, it was found that conditional propositions have a much higher ability to differentiate true from false donors than simple propositions. Compound propositions can misstate the weight of evidence given the propositions strongly in either direction.

## 1. Introduction

When forensic DNA testing reveals a concordance between a crime scene sample and a person of interest’s (POI) DNA profile, it is necessary to provide a statistic to evaluate the strength of the correspondence or the weight of the evidence.

The likelihood ratio (*LR*) is acknowledged as the most powerful and relevant statistic used to calculate the weight of DNA evidence and is recommended by the DNA commission of the International Society of Forensic Genetics (ISFG) in forensic DNA mixture interpretation [1]. 

The *LR* is a ratio of two conditional probabilities, probability densities, or numbers proportional to them. The *LR* is not exclusively used for the interpretation of forensic DNA evidence. It is used to assign the weight of evidence for other forensic evidence and used in many other situations in statistics. It follows from Bayes’ theorem where the odds form is:$$\frac{\mathrm{Pr}\left(H_{p}|E,I\right)}{\mathrm{Pr}\left(H_{d}|E,I\right)}=\frac{\mathrm{Pr}\left(E|H_{p},I\right)}{\mathrm{Pr}\left(E|H_{d},I\right)}\times \frac{\mathrm{Pr}\left(H_{p}|I\right)}{\mathrm{Pr}\left(H_{d}|I\right)}$$
where *E* represents the evidence, *I* represents relevant background information, and *H_p_* and *H_d_* (or *H_a_*) represent alternate hypotheses or propositions. Bayes’ theorem follows directly from the laws of probability and can be expressed in words as follows:

Posterior odds = likelihood ratio × prior odds.

An *LR* greater than one means the DNA evidence supports the proposition given in the numerator. An *LR* less than one means the evidence supports the alternate proposition given in the denominator. In forensic casework, the *LR* in Bayes’ theorem is typically written as shown above, with the probability of the evidence given the prosecution hypothesis forming the numerator and the probability of the evidence given the defence hypothesis as the denominator.

The prosecution proposition (*H_p_*) is generally known and straightforward to apply, especially when only one POI is being considered. The defence are under no requirement to offer a proposition, and often they do not. If the defence proposition is available, then that should be selected. If not, a sensible ‘alternate’ proposition consistent with exoneration should be chosen. Hence, the use of *H_a_* for an alternate proposition can be a preferred descriptor.

There is a well-established hierarchy of propositions that are informed by the evidence being assessed. The original three levels within the hierarchy are offence, activity, and source-level propositions [2]. Forensic DNA evidence is typically evaluated at the sub-source or sub-sub-source level within the hierarchy [3,4]. Within this paper, we discuss *LR*s assigned using sub-source level proposition sets. Below, we give an example of a sub-source set of propositions for a two-person mixed DNA profile considering one POI as a contributor (set one):

Set one, the simple proposition pair, sub-source propositions (*LR* for a single POI, no conditioning):

*H_p_*: The DNA originated from the POI and one unknown individual, unrelated to the POI

*H_a_*: The DNA originated from two unknown individuals, unrelated to the POI or each other

The propositions assigned in a case should be mutually exclusive, address the issue of interest and be close to exhaustive in that they take account of relevant case information and ensure no reasonable consideration is omitted [4,5]. The propositions considered must be plausible or sensible within the known framework of circumstances. The use of non-sensible propositions can lead to misleading *LR*s [6,7].

If one is transparent about the information that has been used to form the propositions and willing to consider a re-evaluation of the findings given different propositions, should the information change, then this approach is robust.

A simple proposition pair is where no more than one POI considered within *H_p_* is replaced with an unknown individual within *H_a_*. Proposition set one above is an example of a simple proposition pair.

In the case of circumstances where there is more than one POI, there are multiple propositions that may be considered both under *H_p_* and *H_a_*. Consider a two-person mixture where two POI both give inclusionary *LR*s using a simple proposition pair. In this case, it is prudent to test whether these POI could explain the profile when considered together. This could be undertaken using a compound proposition pair, defined as one where more than one POI within *H_p_* is replaced with unknown donors in *H_a_* ([8], hereafter the ASB (American Standards Board) draft standard and see also [9,10]).

Set two, the compound proposition pair, sub-source propositions (*LR* for all POI together, no conditioning):

*H_p_*: The DNA originated from POI_1_ and POI_2_

*H_a_*: The DNA originated from two unknown individuals, unrelated to either POI or each other

Although this proposition pair is highly effective in assessing whether both POI could be donors together, reported without the simple *LR*s for each individual, it can appear to greatly overstate the weight against a POI who gives a small inclusionary or uninformative *LR* when considered individually but who is carried in the compound *LR* by the much stronger other donors to the mixture.

Another form of proposition pair assumes the contribution of all POIs under *H_p_* and *all but one POI* under the alternate proposition. We cannot find a definition of this proposition pair in the ASB draft standard [8], although this appears to come under clause 4.5.b, where they are described as a variant of the simple proposition pair. We will term these conditional proposition pairs. If the contribution of all POIs is supported by the observations, then the *LR* for such a conditional proposition pair is a good approximation of the exhaustive *LR*, as described by Buckleton et al. [7] (their Equations (7a) and (7b)).

Set three, the conditional pair, considering POI 1 for a four-person mixture (*LR* for a single POI, uses conditioning profiles):

*Hp*: The DNA originated from POI1, POI2, POI3, and POI4

H_a_: The DNA originated from POI_2_, POI_3_, POI_4_, and one other individual, unrelated to POI_1_, POI_2_, POI_3_, and POI_4_

Three additional conditional *LR_s_* would subsequently be assigned considering POI_2_, POI_3_, and POI_4_. This isolates the evidence for the contribution of each POI in turn. Note that there are other possible combinations of conditional propositions when considering mixtures of more than two individuals. For example, conditioning on only one or two known contributors within a four-person mixture. These partial conditioned *LR*s are not calculated within this paper but are explored by Duke et al. [9] (see, for example, the study’s Table 4).

Given these types of scenarios, the assignment of a compound *LR* is advised by the ASB draft standard [8]. However, as these *LR*s may overinflate the evidence, they advise that the *LR*s derived from simple proposition pairs are the ones reported and not the compound *LR* unless this is exclusionary.

Bright and Coble [11] report that for individuals who are well-represented in the mixture, the logarithm of the compound *LR* is approximately the sum of the logarithm of the *LR*s for each of the known contributors considering simple proposition pairs. This is only approximately true and then only for true donors. Within this research, we investigate the behaviour of *LR*s assigned for known and non-contributors to a set of mixed DNA profiles using compound, conditional, and simple proposition pairs. We demonstrate that compound likelihood ratios can be obtained as the product of conditional likelihood ratios. We also demonstrate that, on average, conditional *LR*s result in higher *LR*s for a true donor and more exclusionary *LR*s for non-contributors than their equivalents using simple proposition sets.

## 2. Materials and Methods

### 2.1. Data

Two sets of mixed GlobalFiler™ DNA samples from two different laboratories (termed Lab A and Lab B) were amplified. The data comprised thirty-two mixtures (two laboratories × four NOCs × four mixtures) consisting of four each of N = 2, N = 3, N = 4, and N = 5 contributors. Samples from Lab A were amplified using 28 PCR cycles and analysed on a 3500 Genetic Analyser with 1.2 kV, 20 s injection parameters and samples from Lab B were amplified using 29 PCR cycles and analysed on a 3500 Genetic Analyser with 1.2 kV, 24 s injection parameters. All profiles were analysed in GeneMapper™ ID-X V1.6 with analytical thresholds (AT) of 125 rfu and 100 rfu for Lab A and B, respectively.

### 2.2. Interpretation and LR Assignment

Profiles were interpreted using STRmix™ (https://strmix.com/, accessed on 1 September 2022 [12,13]) V2.8 (Lab A) and V2.9.1 (Lab B), assuming the apparent number of contributors, which also equalled the experimental number of contributors. A summary of the STRmix™ assigned template and mixture proportion and the experimental design for each mixture is given in Appendix A Table A1. These mixtures cover a broad range of template amounts, number of contributors, and mixture proportions and are representative of DNA profiles typically encountered in casework.

*LR*s were assigned in STRmix™ for each known contributor using a set of simple propositions. Following the nomenclature of Slooten [14] for a two-person mixture, the *LR* assigned for POI_1_ using a simple proposition set is:$$LR_{1u/uu}=\frac{L_{1,u}}{L_{u,u}}$$
where, for example, $L_{1,u}=\mathrm{Pr}(E|H_{1,u},I)$ and *H*_1,*u*_ is the proposition that POI_1_ and an unknown person unrelated to POI_1_ are the donors.

Compound *LR*s were assigned in STRmix™ for each mixture of the type $LR_{12/uu}=\frac{L_{1,2}}{L_{u,u}}$. Compound proposition pairs that gave an *LR* of 0 (exclusion) for true donors were re-deconvoluted using increased numbers of burn-in and post-burn-in accepts (×10 or ×100 compared to defaults).

In addition, each profile was interpreted in STRmix™ V2.9.1 N times (where N is the number of contributors assigned to each mixture), each allowing the approximation to the exhaustive *LR*s to be assigned of the type $LR_{12/2u}=\frac{L_{1,2}}{L_{2,u}}$ and $LR_{12/1u}=\frac{L_{1,2}}{L_{1,u}}$. These are termed conditional *LR*s.

All *LR*s were assigned using the NIST 1036 Caucasian allele frequencies [13] and F_ST_ = 0.01.

### 2.3. Compound LR Derivation Using DBLR™

Conditional and simple *LR*s were assigned in DBLR™ (https://strmix.com/dblr/, accessed on 1 September 2022) for the 32 mixtures from Lab A and B. Sub-source *LR*s were assigned conditioning on the presence of each POI in turn. The conditional *LR*s were built sequentially depending on the number of contributors in the mixture. A derivation of the compound *LR* using conditional and simple *LR*s is given in Appendix B. For example, for a two-person mixture with two POIs, the compound *LR* can be written as the product of a conditional *LR* and a simple *LR*:$$LR_{12/uu}=\frac{L_{1,2}}{L_{u,u}}=\frac{L_{1,2}|L_{1,u}}{L_{u,u}|L_{1,u}}=LR_{12/1u}LR_{1u/uu}$$
where *L*_1,2_ is the likelihood of POI_1_ and POI_2_ both being contributors, *L*_1,*u*_ is the likelihood of POI_1_ and one unknown, and *L_u_*_,*u*_ is the likelihood of two unknown contributors to the two-person mixture. $LR_{12/1u}$ relates to the question: what is the likelihood of POI_2_ also being present in the mixture, given POI_1_ is present?

For a three-person mixture, we consider the following:$$LR_{123/uuu}=\frac{L_{1,2,3}}{L_{u,u,u}}=\frac{L_{1,2,3}|L_{1,2,u}}{L_{u,u,u}|L_{1,2,u}}=LR_{123/12u}LR_{12u/1uu}LR_{1uu/uuu}$$

Conditional *LR*s in DBLR™ differ from those in the STRmix™ because the POI can be assumed to be present without the need to undertake a separate deconvolution. Conditioning on the presence of a contributor in STRmix™ is only possible during deconvolution. Hence, if a deconvolution was first undertaken without conditioning, then a second deconvolution would need to be performed with conditioning before a conditional *LR* could be assigned. In the DBLR™ software, however, it is possible to assume the presence of a contributor even if the deconvolution was undertaken without conditioning. This makes it possible to use a single deconvolution to evaluate various conditional likelihood ratios. A factorisation of a compound likelihood ratio as a product of conditional likelihood ratios is exact because the weights for the genotype sets remain the same between the *LR* assignments. If STRmix™ is used to assign the conditional likelihood ratios, then these factorisations will hold only approximately as the weights for the genotype sets change.

*LR*s were assigned using the NIST 1036 Caucasian allele frequencies [13] and F_ST_ = 0.01.

### 2.4. Non-Contributor LRs

#### 2.4.1. Compound Propositions

In addition to the *LR*s for known contributors, *LR*s for non-contributors were assigned. Three mixtures from each lab were selected: a three-, a four- and a five-person mixture where the compound propositions using all known donors produced a log_10_(*LR*) exceeding the sum of the sub-source log_10_(*LR*) for each individual donor. Two non-contributor genotypes were selected for each of the six mixtures. These non-contributors were either selected from a set of random donors where they resulted in inclusionary *LR*s using a simple proposition set or were constructed using genotypes from the known donor profiles. The inclusionary *LR*s ranged from two to over 38 million.

Compound *LR*s were assigned for each mixture where the non-contributor replaced a true donor under *H_p_*. This was repeated with the non-donor replacing each true donor in each mixture set. For example, for the four-person mixtures, four compound *LR* calculations were undertaken where *H_p_* was considering:Donor 1, Donor 2, Donor 3, Non-donor ADonor 1, Donor 2, Non-donor A, Donor 4Donor 1, Non-donor A, Donor 3, Donor 4Non-donor A, Donor 2, Donor 3, Donor 4.

#### 2.4.2. High-Risk Database, Simple and Conditional Propositions

In addition, two high-risk databases of non-contributors were generated by randomly sampling alleles from the known contributors to each of the mixtures for each laboratory. In this manner, 1000 profiles were generated. *LR*s were assigned for each of the 32 mixtures using a simple and conditional proposition set. The conditional proposition set is conditioned on N-1 known donors. The simple proposition set considered the POI (the non-contributor from the high-risk database) under H_p_ and an unknown under H_a_. The two sets of 1000 LRs were calculated within STRmix™ using the NIST 1036 Caucasian allele frequencies [13] and F_ST_ = 0.01.

Approximate compound *LR*s were additionally assigned for non-contributors who gave non-exclusions (*LR* ≠ 0) when using the conditional proposition set. These compound *LR*s were approximated by summing the log_10_(*LR*) for one known contributor using the simple proposition set and conditional log_10_(*LR*)s, as shown in Appendix B.

## 3. Results

For each mixture, the compound log_10_*LR* assigned in STRmix™ was the same as the sum of the conditional log_10_*LR*s and one simple log_10_*LR* assigned in DBLR™ (the log_10_(*LR*) was compared to six decimal places). This is the expected result.

A summary of the sub-source *LR*s assigned using the simple proposition set and compound proposition set for the Lab A and Lab B mixtures is given in Figure 1. *LR*s using simple proposition sets and the true donors are given as stacked columns where the *LR* for each contributor is given as a different colour. The compound *LR* is given for each mixture as a red asterisk.

Exclusions (*LR* = 0) were obtained for nine of the 32 mixtures using a compound proposition set and the true donors. These included one four-person mixture and all eight five-person mixtures. This is not unexpected when there are multiple unknown contributors. The sample space is so vast that it can be inadequately sampled by the number of default accepts. These profiles were re-interpreted in STRmix™ with ×10 or ×100 the default accepts (100,000 or 1,000,000 burn-in and 500,000 or 5,000,000 post-burn-in accepts) per chain to better explore the probability space in the deconvolution (see Appendix A). Following reinterpretation, compound *LR*s > 1 were assigned for all nine mixtures. These are the results shown in Figure 1.

Inspection of Figure 1 shows that the compound log_10_(*LR*) were larger than the sum of the individual log_10_(*LR*)s using the simple proposition set for each known contributor for all but one sample. This is more pronounced for the high-order mixtures (N = 3 and greater). This is an overrepresentation of the weight of evidence against each individual contributor.

The five-person mixture (Lab B, sample number 3), designed with donor ratios of 10:2:2:1:1 and with a 100 pg template for the lowest contributors interpreted using ×100 accepts resulted in a compound log_10_(*LR*) that was less than the sum of the individual log_10_(*LR*)s (52.26 versus 57.47). The mixture proportions assigned by STRmix™ were 64%, 16%, 11%, 8% and 1%. The contributor position with the highest *LR* for two of the contributors to this mixture using simple proposition sets differed from the contributor order they aligned with for the compound *LR*. The sub-sub-source *LR* for one contributor was approximately 20 times lower in its compound *LR* position. The sub-sub-source *LR* for the other contributor was around 17 orders of magnitude lower. This contributor best aligned in the third contributor position using simple propositions with an approximate mixture proportion of 11% but was aligned as the trace fifth contributor with an approximate mixture proportion of 1% using the compound proposition set. This individual is one of the two lowest template donors. Their alignment in the third contributor position using simple propositions is likely due to the presence of a D2S1338 18.3 peak not originating from any actual donor and likely drop-in, which is favoured as an allele for the fifth contributor, and also given the amount of allele sharing between donors. The sum of the individual log_10_(*LR*) for each donor with simple propositions when in their experimentally designed contributor positions was 40.67.

### 3.1. Conditional LRs

A plot of the log_10_(*LR*) assigned for the true donors to the 32 mixtures using the simple proposition set (per contributor) versus the conditional log_10_*LRs* (alternatively described as Slooten and Buckleton et al.’s approximation to the exhaustive (*LR*)) is given in the top pane of Figure 2. The *LR*s assigned given conditional propositions were larger than the *LR*s assigned using simple proposition sets for the same POI for all but one comparison. This was the five-person mixture from Lab B, sample number 3, discussed above. The data points for samples on the *x* = *y* line are for mixtures that were fully or close to fully resolved, and conditioning did not add any extra information to the interpretation.

A plot of the log_10_(*LR*) assigned for the mixtures using compound propositions versus log_10_(*LR*)s for the conditional propositions is given in the bottom pane of Figure 2**.** The *LR*s assigned given conditional propositions were smaller than the *LR*s assigned using compound proposition sets. The data points at [~28, ~0] and [~28, ~27] and indicated as filled data points in Figure 2 are considering two different POIs contributing to the same mixture. The major is (almost) fully resolved, whereas the minor is very ambiguous. The major carries the minor in the log_10_(*LR*) considering compound propositions. When conditioning on the major (in the approximation of exhaustive propositions), no information is gained in relation to the minor’s genotype. Vice versa, when conditioning on the minor, no information is gained in relation to the major’s genotype.

### 3.2. Non-Contributor Tests

#### 3.2.1. Compound Propositions

The twelve non-contributors (two for each of the six mixtures tested in Section 2.4.1) that had previously given inclusionary *LR*s using simple proposition sets resulted in exclusions (*LR* = 0) when using compound propositions, where they replaced, one by one, each of the true donors in the proposition.

#### 3.2.2. High-Risk Database, Simple and Conditional Propositions

A plot of log_10_(*LR*) given a simple proposition set versus the template assigned in STRmix™ (in rfu) for the high-risk database of non-contributors is given in Figure 3. Overall, 56% of comparisons were exclusions (*LR* = 0) and are plotted around log_10_(*LR*) = −40 in Figure 2.

A plot of log_10_(*LR*) given a conditional proposition set versus the template assigned in STRmix™ (in rfu) for the high-risk database of 1000 non-contributors is given in Figure 4. The conditioned individual(s) was a known donor, and the POI was a database individual. Over 99% of comparisons resulted in *LR* = 0.

Compound log_10_(*LR*) values for non-contributors within the high-risk database, which resulted in *LR* > 0 when assigned using a conditional proposition set, are plotted against the corresponding conditional log_10_(*LR*) values in Figure 5. The compound *LR* is always greater than the conditional *LR* for the non-donors.

## 4. Discussion

The conditional *LR* showed that, on average, the *LR* assigned to true donors was larger than the *LR* assigned using simple proposition sets for the same POI. This is because conditioning on another true donor adds information to the interpretation allowing for better resolution of the remaining genotypes. This is the known effect of conditional *LR*s. The data points on or about the line of equality in Figure 2 (top pane) are profiles that were fully resolved (or close to fully resolved), where conditioning on a contributor did not add extra information to the interpretation. The conditional *LR* was always lower than (or equal to) the *LR* using compound propositions (Figure 2 bottom pane).

The rate of adventitious matches for high-risk non-contributors created by sampling alleles from known contributors was significantly higher when using simple proposition sets compared with conditional proposition sets (Figure 3 versus Figure 4). Conditional *LR*s have an increased power to differentiate between true and false donors. Conditioning on a true donor should increase *LR*s for other true donors (as demonstrated in [15]) and lower them for false donors. This is demonstrated again within this work. The high-risk non-contributors represent a ‘worst case’ scenario not typically encountered in casework other than when mixtures of relatives are involved.

In relation to simple proposition sets, Slooten states [14], “The hypotheses for $LR_{1u/uu}$ only use the person of interest under investigation here. It may seem at first sight as an unbiased way to present the evidence, not using any other POI whose contribution is also disputed as assumed contributors. But it is easily overlooked that in fact one then assumes that the other POI did not contribute, and that this assumption is not at all supported by the data.” The simple proposition under *H_p_* might also not represent the most logical scenario for the prosecution given the case circumstances.

The assignment of a compound *LR* is a natural extension if multiple POIs give inclusionary statistics when using simple proposition sets. We have shown that the logarithm of the compound *LR* is the sum of conditional log_10_(*LR*) and a simple log_10_(*LR*) for the individual contributors. However, the compound *LR* is only useful as a test of whether two or more POI can both be donors. In the overwhelming majority of cases, they are an inappropriate expression of the weight of evidence for any individual donor and may be too high or too low. Compound proposition sets have a higher chance of both false inclusionary support (non-donor carried by strong *LR*s of other donors), as shown in Figure 5 and false exclusionary support (*LR* = 0 due to the vast sampling space and computing limitations).

In general, if multiple POIs can be included in a mixture individually, and the ground truth is that all POIs have contributed, we expect the compound log_10_(*LR*) to be greater than the sum of the log_10_(*LR*)s assigned using simple proposition sets (Figure 1). Mixtures with the greatest ambiguity (or least well resolved) will typically have the greatest difference between the compound log_10_(*LR*) and the sum of the individual simple log_10_(*LR*)s (refer to Appendix B). This is because, in the compound *LR*, the *LR*s for the individual contributors (for example, *LR*_1_ and *LR*_2_ for a two-person mixture) are not independent. Conditioning on a POI adds information to the interpretation, reducing the number of genotype combinations possible for the remaining contributor/s.

Fully resolved mixtures are a special case where the compound log_10_(*LR*), the sum of conditional log_10_(*LR*)s, and the sum of the simple log_10_(*LR*) for each true donor POI will all be equal, as long as sub-sub-source propositions are considered. This is because when the mixture is fully resolved in the compound *LR* calculation $LR_{1u/uu}$ and $LR_{2u/uu}$ are now independent, i.e., conditioning on a POI being present does not add any extra information to the calculation.

We have demonstrated that, for some samples with a high number of contributors, the compound *LR* is zero even though for each true donor POI the simple *LR* is inclusionary. In these samples, the genotype combinations of all true donor contributors individually were accepted at least once across the posterior burn-in iterations, but the genotype combination explaining all true donor contributors in combination was not accepted within one iteration. This is not unexpected and arises because the sample space is vast. In these cases, we recommend the use of extended MCMC accepts within the interpretation. This allowed for more time to explore the sample space and was also a finding of Duke et al. [9].

Where it is necessary to determine if multiple POIs could together be donors to relatively high template complex mixtures comprising four or five contributors, this may require additional MCMC accepts to fully explore the range of possible genotype combinations at each locus. The additional accepts may allow a wider range of genotype combinations to be accepted, thereby preventing an exclusion.

## 5. Conclusions

When assigning LRs in forensic casework, an analyst may have some idea of the most appropriate prosecution proposition but very rarely has knowledge of the most appropriate defence proposition. In the absence of this information, a reasonable set may be selected in a way that maintains the legitimate interests of the defence. This can be informed by case circumstances. An understanding of the performance of the LR under certain proposition sets can also help an analyst make this decision.

It may be worthwhile benchmarking two of the recommendations in the draft ASB standard; recommendations 4.4 and 4.5 [8]. We note that these recommendations are in draft.

Recommendation 4.4: A profile should be assigned as a conditioning profile to a mixture when an individual is identified as an intimate contributor or when it is reasonable to assume their presence based on case-specific information and the associated data supports the assumption. The conditioning profile could be from the complainant, POI, or other individuals, depending on the case scenario. In the published guidelines for setting sub-source propositions [3], the DNA Commission of the International Society for Forensic Genetics define relevant case circumstances as those that “include only the case information that is needed for the formulation of the propositions and for assigning the probabilities of the results”. Buckleton et al. [4] describe forensically relevant case circumstances for a DNA case as “information that will help formulate the appropriate alternative, determine the number(s) of contributors, and select the relevant population”. They do not consider “information such as prior conviction, motive, presence of other types of evidence, or a confession as relevant forensic information”. As much relevant case information should be gathered as practical before formulating the propositions.

The conclusions of this work suggest that this recommendation should be greatly strengthened. We reprise Slooten’s insightful comment that not conditioning is also an assumption [14]: That the profile being considered for conditioning is not a donor and that this is not at all supported by the data. It is very tempting to feel that not assuming is somehow safe or conservative. But the choice is between assuming that the conditioning profile is, or is not, a donor. If the data support the presence of this profile, it can be very detrimental not to assume their presence. This is because of the much-enhanced ability to differentiate true from false donors when conditioning is applied. A useful way through this issue is to use the approximation to the exhaustive *LR*. This enables a balanced approach that assumes that the conditioning profile either is or is not a donor. However, in the event that two or more donors cannot both or all be donors (or that the compound *LR* is much less than 1), it is still necessary to state this explicitly.

Recommendation 4.5: The analysis should separate the propositions into their simplified constituents (i.e., simple proposition pairs—recall that ASB describes both simple and conditional propositions as simple) when an *LR* favouring *H_p_* has resulted from a compound proposition pair incorporating multiple POIs under *H_p_* and none of the POIs under *H_a_*, in order to establish the weighting and the consequent probative value of the evidence per contributor under *H_p_*.

The conclusions of this work very strongly support this statement. Compound proposition pairs can misrepresent the weight of the evidence against an individual strongly in either direction. This work strongly favours the use of conditional proposition pairs rather than simple proposition pairs whenever the data support the presence of an individual as the conditioning profile since this increases the ability to differentiate true from false donors.

We have demonstrated by calculating conditional *LR*s in DBLR™ that the compound *LR* can be obtained as a product of simple and conditional *LR*s. This is also approximately true using *LR*s produced by STRmix™. The use of conditional *LR*s, described as an approximation to the exhaustive *LR* by Buckleton et al. [7], resulted in higher *LR*s for the known contributors and lower *LR*s for the non-donors than when using a simple proposition pair. This statistic makes the best use of the DNA profiling information.

## Figures and Tables

**Figure 1 genes-14-00714-f001:**
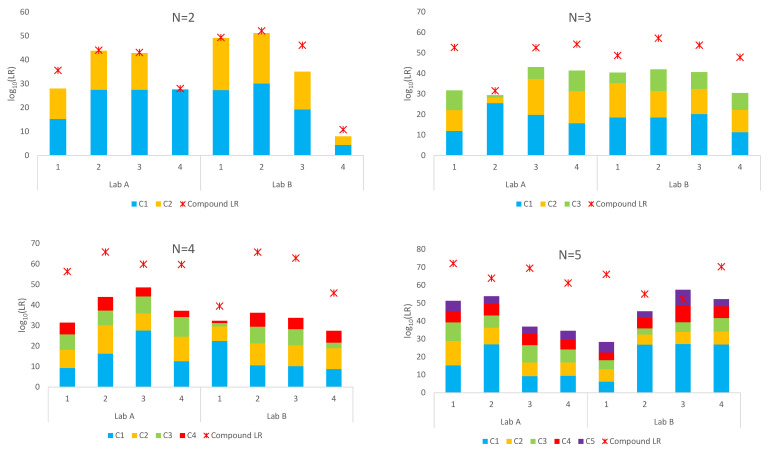
*LR*s for mixtures (N = 2 through N = 5) for Lab A and Lab B using simple proposition sets for each contributor and a compound proposition set considering all known contributors.

**Figure 2 genes-14-00714-f002:**
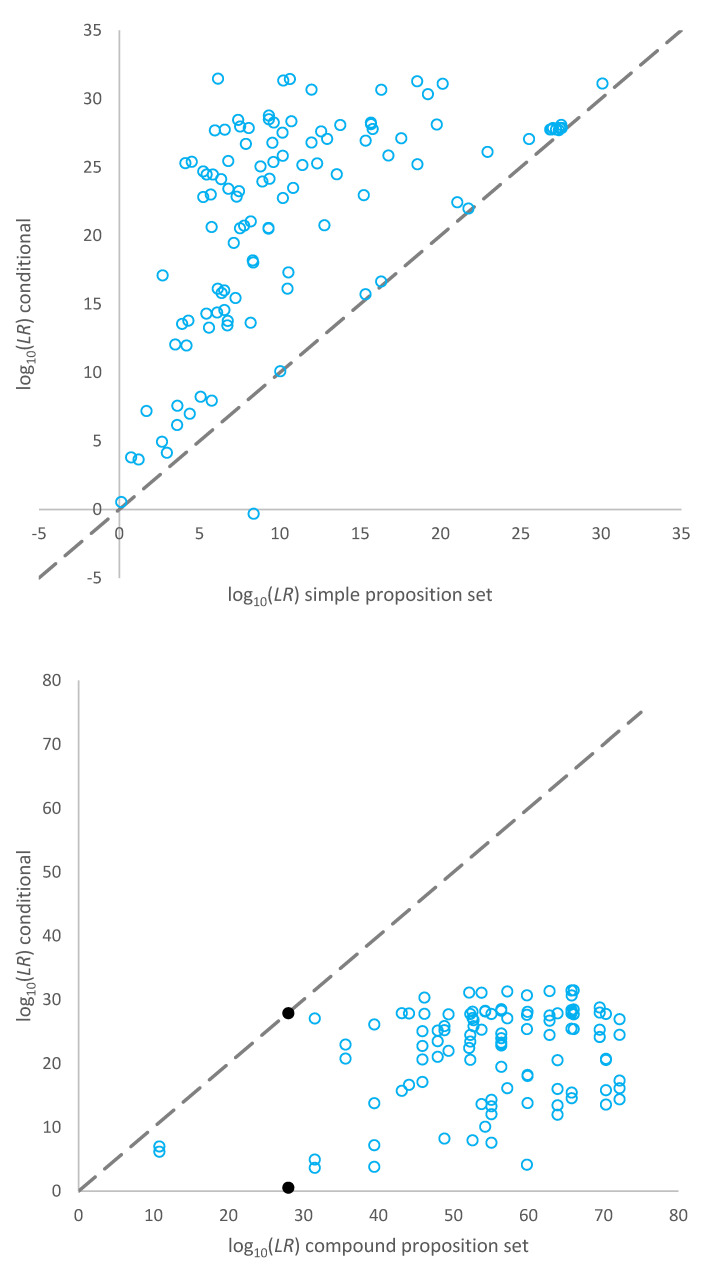
Top pane: Plot of log_10_(*LR*)s assigned for each known contributor using a simple proposition set versus conditional propositions (an approximation to the exhaustive log_10_(*LR*)). Bottom pane: Plot of log_10_(*LR*)s assigned for compound propositions versus an approximation to the conditional log_10_(*LR*) considering each POI in turn.

**Figure 3 genes-14-00714-f003:**
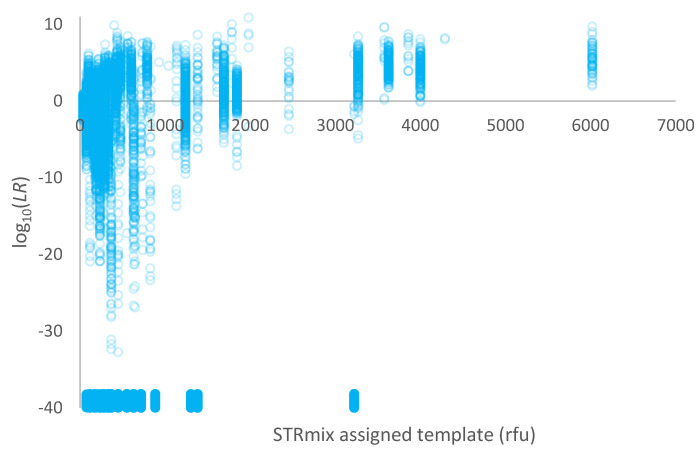
Plot of log_10_(*LR*) given a simple proposition set versus the template assigned in STRmix™ (in rfu) for 1000 non-contributors within a high-risk database. Exclusions (*LR* = 0) are plotted as log_10_(*LR*) = −40 and have been jittered along the *y*-axis to better display the points.

**Figure 4 genes-14-00714-f004:**
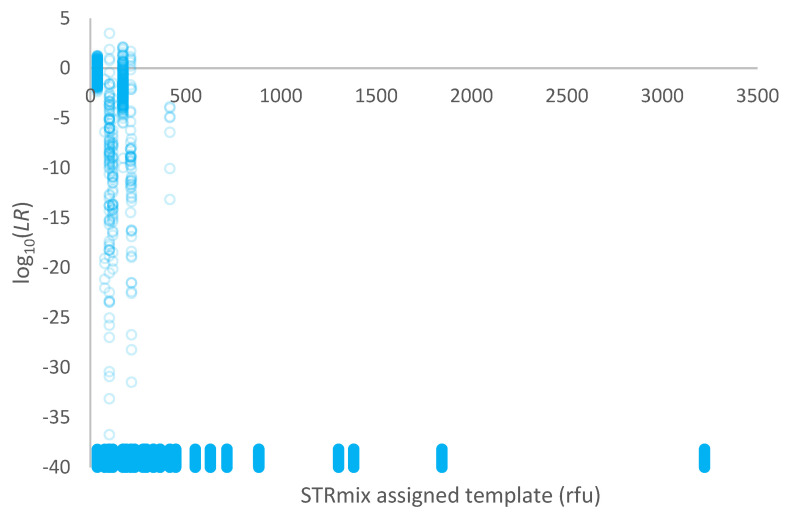
Plot of log_10_(*LR*) given a conditional proposition set versus the template assigned in STRmix™ (in rfu) for 1000 non-contributors within a high-risk database. Exclusions (*LR* = 0) are plotted as log_10_(*LR*) = −40 and have been jittered along the *y*-axis to better display the points.

**Figure 5 genes-14-00714-f005:**
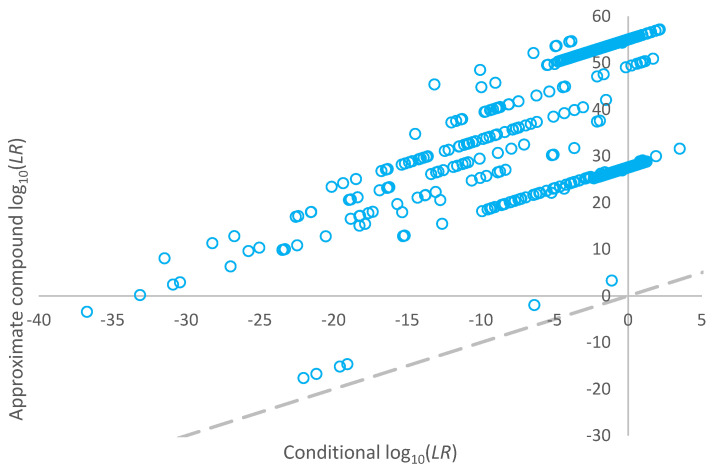
Plot of log_10_(*LR*) given a conditional proposition set versus the approximate compound log_10_(*LR*) for non-contributors within the high-risk database which resulted in *LR* > 0 when assigned using a conditional proposition set.

## Data Availability

Data is unavailable due to privacy reasons.

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
