# Peer review of "An Investigation into Compound Likelihood Ratios for Forensic DNA Mixtures"

_genes, 2023, doi:10.3390/genes14030714_

Round 1

Reviewer 1 Report

I would better be able to understand the results if the "error! Reference source not found" was corrected. There are several spots in the results and discussion where this occurs. I assume these spots are referencing a particular figure. 

Author Response

Thanks for your review and comments. Apologies for referencing issues relating to the different figures within the text. I have updated the manuscript and corrected the 'errors' so that the text points to the releavant Figure in which the data is presented.

Reviewer 2 Report

****There are many instances of Error! Refer-ence source not found.. throughout the manuscript. This should be corrected. ****

1 - Introduction

Introduction is very exhaustive; it gives an appropriate context and dives into the underpinning theories. It then appropriately explains what simple propositions are, compound propositions or conditional propositions and what is the expected behavior of an LR calculated using these sets of propositions.

Scope of the work is well defined, and a short summary of the conclusion is included.

Material and methods

*2.1 – I would add a short description of the input DNA used and expected mixture ratio that is not calculated by STRMix.

*2.2 – Was the number of contributors assessed by more then one analyst? Was the NOC established without any prior knowledge of the ground truth? Did the mixtures we construct to ensure that they would result in a definitive NOC: low template DNA mixture can lead to a range of possible NOC (ex.3-4).

2.2- Was a model generated using the validation parameters of Lab A and Lab B?

3 – Results

Results are well described, and the figures are high quality. Reference to appendix are used to add additional information.

4- discussion

«We have demonstrated that, for some samples at high number of contributors, the compound LR is zero even though for each true donor POI the simple LR is inclusionary. In these samples, the genotype combinations of all true donor contributors individually were accepted at least once across the posterior burn-in iterations but the genotype combination explaining all true donor contributors in combination was not accepted within the one iteration. This is not unexpected and arises because the sample space is vast.

Where it is necessary to determine if multiple POIs could together be donors to relatively high template complex mixtures, comprising 4 or 5 contributors, this may require additional MCMC accepts to fully explore the range of possible genotype combinations at each locus. The additional accepts may allow a wider range of genotype combinations to be accepted thus preventing an exclusion. »

This is an interesting finding; I would suggest that a brief discussion is included as this relates to casework. Should an analyst always re-deconvolute using an increased numbers of burn-in and post burn-in accepts when dealing with multiple inclusion in 4 or 5 contributors’ mixtures to allow the exploration of a wider space? Is there a risk that an iteration be accepted but not representative of the ground truth?  The authors should try to provide a practical user with some guidance on this issue.

 «Recommendation 4.4: A profile should be assigned as a conditioning profile to a mixture when an individual is identified as an intimate contributor, or when it is reasonable to assume their presence based on case specific information, and the associated data supports the assumption. The conditioning profile could be from the complainant, POI, or other individual depending on the case scenario. »

I would suggest to the authors that they give some guidance regarding case specific information that should be used to condition a profile. Can this result into a bias analysis of fact? To what extend should an analyst gather information before conduction its analysis?

Author Response

Thanks for your review and comments.

I have amended the manuscript so that the relevant figure numbers are displayed where reference errors were present.

R2. Point 2.1. The experimental design for each mixture in the study is now shown within Table A1 in the Appendix.

R2. Point2.2. a) At both laboratories (A & B), the NOC was assigned by a trained analyst, unaware of the mixture design or set up. The mixtures selected for this study were ones where the assigned NOC by the analyst mirrored the experimental design.

  1. b) The parameters applied to the deconvolutions aligned with the values determined for each laboratory’s internal validation study.

R2. Point 4. We have added the following to the manuscript to help guide users of the software.

 “In these cases, we recommend the use of extended MCMC accepts within the interpretation. This allows for more time to explore the sample space and was also a finding of Duke et al.”

R2. Re OSAC recommendation 4.4. We have added the following paragraph.

“In the published guidelines for setting sub-source propositions, the DNA Commission of the International Society for Forensic Genetics define relevant case circumstances as those that "include only the case information that is needed for the formulation of the propositions and for assigning the probabilities of the results". Buckleton et al. (Helping formulate propositions in forensic DNA analysis) describe forensically relevant case circumstances for a DNA case as "information that will help formulate the appropriate alternative, determine the number(s) of contributors, and select the relevant population". They do not consider "information such as prior conviction, motive, presence of other types of evidence, or a confession as relevant forensic information". As much relevant case information should be gathered as practical prior to formulation of the propositions.”